# Socio-economic gradients in hypertension and diabetes management amid the COVID-19 pandemic in India

**Toshiaki Aizawa** *

Graduate School of Economics and Business, Hokkaido University, Hokkaido, Japan

* toshiaki.aizawa@econ.hokudai.ac.jp

## Abstract

This study examines socio-economic inequalities in the prevalence and treatment of hypertension and diabetes among adults in India, utilising data from the National Family Health Survey (NFHS) collected before and during the COVID-19 pandemic. Disparities associated with individual demographic and socio-economic characteristics are measured, with the level of inequality quantified using the dissimilarity index and contributing factors analysed through decomposition analysis. The results reveal significant socio-economic gradients, with wealthier individuals more likely to have elevated blood pressure and blood glucose levels and to treat them. Socio-economic gradients in treatment are even steeper among middle-aged groups during the pandemic. These wealth- and education-related disparities become more pronounced with age. This study highlights the need for targeted interventions and policies to address socio-economic disparities in access to essential care for socio-economically disadvantaged populations.

## Introduction

### Treatment for non-communicable diseases during the COVID-19 pandemic

High blood pressure and diabetes are among the most well-known causes of life-threatening complications, along with heart attacks, strokes and kidney failure [1, 2]. In the case of hypertension, over the past 30 years, the number of adults aged 30–79 with hypertension doubled, now reaching an estimated 1.3 billion worldwide [3]. They seldom cause symptoms at an early stage, and people are rarely aware of the possible risk, thus tending to postpone medical examination and/or treatment. Moreover, under-treatment over a long period can have considerable negative social and economic impacts on a country's welfare, as well as lead to high medical costs for patients and their families [4, 5]. People with chronic conditions should receive diagnosis and manage them before it is too late. However, in low- and middle-income countries, which typically have weak health systems, the proportion of people who do not receive adequate treatment is high [3]. In 2019, the probability of premature mortality triggered by hypertension in India was estimated to be 25% for males and 19% for females, corresponding to 1,451,000 and 1,116,000 deaths, respectively [3].

**Data Availability Statement:** Third party data was obtained for this study from The DHS Program (https://dhsprogram.com/). Data may be requested from The DHS Program after creating an account and submitting a concept note. More access

information can be found on The DHS Program website (https://dhsprogram.com/data/Access-Instructions.cfm). The data set is openly available upon permission from the MEASURE DHS website (https://www.dhsprogram.com/data/available-datasets.cfm). The authors confirm that interested researchers would be able to access these data in the same manner as the authors. The authors also confirm that they had no special access privileges that others would not have.

**Funding:** This study is supported by research grants from JSPS Grant-in-Aid for Scientific Research (JP24K1634504). The founders had no role in study design, data collection and analysis, decision to publish, or preparation of the manuscript.

**Competing interests:** The authors have declared that no competing interests exist.

India has witnessed an increase in the prevalence of non-communicable diseases (NCDs) [6, 7]. India's rapid urbanisation, aging population, and obesogenic environments are key drivers of the increasing prevalence of hypertension and diabetes. These conditions rise sharply with age, particularly in middle-aged and older adults, and are strongly linked to urban living and higher household wealth [8]. However, prevalence remains alarmingly high even among the poorest rural households, especially for individuals over 40, 5.9% for diabetes and 30.0% for hypertension [8]. The escalating trend poses a significant public health challenge, underscoring the need for focused interventions in prevention, screening, and treatment.

Amid the COVID-19 pandemic, reductions in healthcare use occurred worldwide [9–13]. People refrained from utilising healthcare for multiple reasons, including fear of infection while visiting a facility, lockdown policies, and stay-at-home orders [14]. The economic impact of the pandemic significantly reduced household incomes, thereby constraining many families' ability to afford healthcare services [15, 16]. Restrictions on the supply side also played a role in the reduced use of healthcare services [17]. Many medical facilities halted elective surgeries, regular check-ups, and non-urgent care, which hindered the management of chronic conditions. Moreover, disruptions in supply chains during the pandemic made it difficult to obtain essential medicines and medical equipment, further restricting access to healthcare, especially for those needing chronic disease treatment. Disruptions to healthcare services were more significant in lower-income countries and reductions in healthcare use were greater among people with less severe illnesses [9, 16].

The consequences of the COVID-19 pandemic on inequalities in health and healthcare utilisation are not immediately evident. While the pandemic and its associated policies led to significant reductions in healthcare service use, it also spurred the adoption of telemedicine [18], which may have helped individuals in rural areas with limited medical resources gain access to healthcare. In rural India, COVID-19-induced healthcare avoidance is influenced by factors such as state of residence, type of healthcare facility, primary occupation, and educational level [19]. Specifically, individuals with education beyond high school, those using government hospitals or clinics, and daily wage workers in agriculture had significantly higher odds of avoiding healthcare services during the pandemic.

## Literature review and aims of this study

The relationship between NCDs and socio-economic status (SES) has been extensively studied worldwide. In advanced countries, research consistently shows that hypertension and diabetes disproportionately affect the poor [20]. Specifically, 42 out of 50 studies on advanced countries report a correlation between lower SES and a higher likelihood of hypertension, with this relationship being particularly strong among women [21]. Higher risks of diabetes are observed among lower SES groups in advanced countries [22, 23]. In low- and middle-income countries, however, the correlation between SES and NCDs is less consistent and exhibits greater variability. For instance, in Jamaica, elevated blood pressure levels were observed in both low- and high-income groups among adults [24]. Conversely, a study in Indonesia revealed socio-economic inequalities in hypertension among women, showing that economically advantaged women have a lower risk of hypertension compared to their less advantaged counterparts [25]. In India, hypertension is significantly more prevalent among people with a lower educational background [26] but recent evidence indicates a positive socio-economic gradient for diabetes in India [27].

Compared to the studies investigating the socio-economic determinants of NCDs, the socio-economic determinants in the management of them are less explored. Existing literature explores the determinants of NCDs care cascade which is consisted of the following: "screening", "awareness of the condition", "on treatment" and "under control" [28–30]. In India, the

hypertension care cascade varies greatly between states, with men, rural residents, individuals from less wealthy households, and unmarried individuals having a significantly higher probability of drop-offs at different stages [31]. Recent studies also report the gradients for the cascade of hypertension treatment in India. A low proportion of individuals with diabetes manage appropriate treatment and control, especially among poorer and less educated groups [32]. Even fewer studies have tried to quantify the socio-economic inequality in the treatment of NCDs and extensively investigated the contributory factors to the observed inequality.

This paper aims to fill a gap in the literature by examining the socio-economic determinants of the presence of hypertension and diabetes, as well as their medication, in India amid the COVID-19 pandemic. This study explicitly quantifies the inequality in the prevalence of hypertension and diabetes, as well as their medication, without relying on self-reported diagnoses. By using objective measurements of blood pressure and blood glucose levels, rather than relying on self-reports, this study can more accurately define cases of proper treatment. Chronic conditions are often asymptomatic in their early stages, and many individuals may remain unaware of their conditions until they become severe. If education is positively correlated with the likelihood of being aware of a disease, the prevalence rate based on self-reports or past history of diagnoses among less educated individuals would be underestimated. This would lead to an underestimated educational gradient in self-reported illness, restricting the validity of the findings. Using measured data with household survey data allows this paper to avoid such a potential problem.

The contributions of this study are twofold. First, it quantifies socioeconomic inequalities in the management of hypertension and diabetes using the dissimilarity index (D-index). Unlike a dichotomous regression approach, which estimates socioeconomic inequality based on the relationship between a binary treatment status and its determinants, the D-index reflects the overall distribution of the predicted probability of receiving treatment across the population. For example, if we are interested in disparities in hypertension treatment between educated and less-educated groups, traditional regression analysis would estimate the average difference in treatment rates between these two groups. While regression analysis might indicate that, on average, less-educated individuals have higher access to treatment than more-educated individuals, it does not reveal where within the distribution these disparities are most pronounced. In other words, regression provides a broad indication of inequality but does not show how treatment is distributed within each group. The dissimilarity index, on the other hand, captures absolute differences in treatment across the entire distribution by measuring the share of individuals who would need to switch treatment status to achieve equality between groups. Visual analysis of the D-index can further reveal whether disparities are pervasive across all levels or concentrated at specific levels, offering a more comprehensive view of the structure of socioeconomic inequality.

Second, the study quantifies socioeconomic gradients and explores the underlying drivers of observed inequality. Understanding the sources of these inequalities is crucial for developing policies aimed at reducing socioeconomic disparities in healthcare utilisation. The findings during the pandemic offer valuable insights for policymakers in India, helping to inform the development of equitable healthcare policies that address the growing burden of NCDs during states of emergency, ensuring that essential treatments remain accessible to all, regardless of socioeconomic status.

## Healthcare system in India

India's public and private hospitals play complementary but distinct roles in healthcare. Public hospitals, operated by the government, provide affordable or free healthcare, particularly

targeting economically disadvantaged populations. These facilities focus on primary care and serve both urban and rural areas. The public healthcare system in India is structured as a three-tier system [33]. At the primary level, care is delivered through sub-centres (serving populations of 3,000–5,000), primary health centres (PHCs) (serving 20,000–30,000 people), and community health centres (CHCs) (serving 80,000–120,000 people). At the secondary level, district hospitals (DHs), equipped with approximately 200 beds, cater to populations of 1–2 million and complement the CHCs in providing secondary care. Tertiary care and advanced medical services are offered by medical colleges. In contrast, private hospitals, primarily located in urban areas, focus on specialised care. These facilities invest heavily in advanced technology and infrastructure, offering higher perceived quality and shorter waiting times for patients willing to pay more than they would in public hospitals.

India's healthcare infrastructure faces significant challenges, marked by stark disparities in the distribution of healthcare facilities, especially between urban and rural areas [34, 35]. Additionally, there is a critical shortage of healthcare professionals, particularly in rural regions. These issues are further exacerbated by financial constraints, with government spending on healthcare amounting to less than 2% of GDP and private sector expenditure just under 3%, both of which are well below the global average of 9.5% [36, 37]. Achieving universal health coverage, a central goal of the Sustainable Development Goals (SDGs), will require sustained investment in healthcare infrastructure and targeted efforts to address socioeconomic inequalities in access to care.

The COVID-19 pandemic had a profound impact on healthcare services in India, with non-COVID-19 medical services, including NCDs screening, suspended from May 2020 onwards [38]. Healthcare resources were diverted to address the pandemic, leading to delays in early detection and treatment of diseases, potentially increasing the burden of advanced-stage conditions [39, 40]. Health ministry staff responsible for NCDs were reassigned to support COVID-19 efforts, further reducing capacity for other essential health services. The restriction of access to non-urgent healthcare during the pandemic disproportionately affected different socioeconomic and age groups [19].

## Methods

### Data

This study employs the latest wave of the National Family Health Survey (NFHS-5), conducted in two phases: from June 17, 2019, to March 23, 2020, and from October 12, 2020, to May 20, 2021, in India. All NFHS surveys were conducted under the stewardship of the Ministry of Health and Family Welfare (MoHFW), Government of India. This paper leverages these sampling characteristics to define two groups: samples before the pandemic and samples during the pandemic. Individuals who were interviewed from June 17, 2019, to January 30, 2020, are classified as samples before the pandemic. The World Health Organization declared the global COVID-19 pandemic on January 31, 2020. Similarly, individuals who were interviewed from October 12, 2020, to May 20, 2021, are classified as samples during the pandemic. Those who were interviewed from January 31, 2020, to March 23, 2020, are not used in this study. During this pandemic period, the Delta variant was first identified in India and subsequently spread worldwide in 2021. Although the NFHS is designed as a nationwide household survey, not all states were surveyed both before and during the pandemic. This study focuses on the states surveyed in both periods to ensure that results are comparable over time. They are Tamil Nadu, Arunachal Pradesh, Jharkhand, Odisha, Chhattisgarh, Madhya Pradesh, Uttarakhand, Haryana, Punjab, Rajasthan, Uttar Pradesh, Puducherry and Dellhi. Through this sample

selection process, all the states in the Western regions were excluded because they were not surveyed during the pandemic. The other regions include at least one state in the sample.

The primary sampling unit comprises two main strata: urban (wards or municipal locations) and rural (villages), which were drawn separately within each state and are proportional to the size of the respective urban and rural areas. Rural areas are defined as having at least one of the following characteristics: (1) fewer than 5,000 residents, (2) population density less than 1,000 per square mile, or (3) at least 25 percent of the adult male population employed in agriculture. Under a multi-stage stratified approach, households were randomly sampled from a list of households in which eligible participants resided. Eligible participants are female participants aged 15–49 years and male participants aged 15–54 years. Selected households were visited by a trained interviewer, who conducted a brief household interview after obtaining written informed consent, completed a roster, and then identified eligible women and men for an individual interview. The survey also collected data on blood pressure and blood glucose from all adults aged 18 years and older living in the same household as eligible participants. Accordingly, all adults aged 18 years and older constitute the sample analysed in this study. For the medication analysis, only those aged 18 years and older who are identified as hypertensive or diabetic are included.

**Hypertension and diabetes.**   In NFHS-5, blood pressure measurements were taken to assess the population prevalence of high blood pressure using the OMRON Blood Pressure Monitor. For consistency, all blood pressure measurements were taken on the respondent's left arm. Blood pressure measurements for each respondent were taken on three separate occasions, with intervals of five minutes between readings [41]. The respondent was asked to avoid eating, smoking, and exercising for 30 minutes before the measurement. This paper follows the WHO definition of hypertension: the condition where the average systolic blood pressure is equal to or greater than 140 millimetres of mercury (*mmHg*) or the average diastolic blood pressure is equal to or greater than 90 *mmHg* [42]. Those who take medication for hypertension are also categorised as hypertensive, regardless of their measured blood pressure values.

Blood glucose levels are used to diagnose diabetes. Random blood glucose was measured using a finger-stick blood specimen for all women and men, using the Accu-Chek Performa glucometer with glucose test strips for blood glucose testing [41]. A high glucose level equal to or greater than 200 *mg/dl* is indicative of diabetes. Those who take prescribed medication for diabetes are also categorised as diabetic, regardless of their measured blood glucose levels.

**Medication.**   For medication against the respective conditions, the following question was asked to those who have ever received a diagnosis of the respective conditions: "Are you currently taking prescribed medication to lower your blood pressure/blood glucose?" Proper medication refers to the circumstances in which individuals with diabetes or hypertension take any prescribed medicine for these conditions. Individuals not suffering from hypertension or diabetes were excluded from the sample when analysing their respective medication, to focus on horizontal inequality in healthcare access among those requiring adequate treatment. Horizontal inequality in healthcare differs from general inequality in that horizontal inequality is the inequality after controlling for healthcare needs. It should be noted that not taking prescribed medicine does not necessarily imply that individuals with these conditions are not managing their health, as regular exercise and improved diets can also mitigate their risks. However, whether or not they are taking medication for the condition provides a more objective measurement of treatment. Consequently, this study uses the current use of medication as a proxy for treatment. Formally, the proper medication for individual *i* are defined as follows:

for $H_i = \{Diabetes_i, Hypertension_i\}$,

$$Proper - medication_i^H = \begin{cases} 1, & \text{if } (H_i, medication_i^H) = (1,1). \\ 0, & \text{if } (H_i, medication_i^H) = (1,0). \\ -, & \text{if } H_i = 0, \end{cases} \qquad (1)$$

where $-$ indicates that the respective outcomes are not determined. Samples with undetermined outcomes will be excluded from the regression analysis.

## Covariates

This study examines the individual characteristics linked to hypertension, diabetes, and their treatment. The choices of these variables are based on previous studies related to socio-economic inequality in health and healthcare in India and other low- and middle-income countries in Asia [43, 44]. Covariates reflecting demographic characteristics include gender and age group binary variables. As a measure of socio-economic status, this study uses household wealth, which is captured by the NFHS through a composite index of relative standards of living derived from indicators of asset ownership, housing characteristics, and water and sanitation facilities. The wealth index in the NFHS is based on principal component analysis [45], developed in collaboration with the World Bank, and has been shown to be a consistent proxy for household income and expenditure [46]. The advantage of using wealth over income is that the former, as a stock of income, is suitable as an indicator for reflecting the long-term living standards of households. The wealth index is divided into quintiles, and the first quintile (poorest) works as a reference category in the quantitative analysis.

Additionally, binary variables for the scheduled castes/tribes and other backward classes are included as indicators of SES. Scheduled castes/tribes are the most socially disadvantaged groups, having endured the greatest burden of social and economic segregation and deprivation within the traditional Hindu caste hierarchy [47]. Scheduled tribes consist of approximately 700 officially recognised groups that historically tend to be geographically isolated, with limited economic and social interactions with the rest of the population [47]. Essentially, they represent the "indigenous" groups in India. Other backward classes are a diverse collection of intermediate castes that, while low in the caste system, rank above the scheduled castes. The remaining classes, including those not identifying as legislatively marginalised, serve as the benchmark category in this analysis.

Educational levels, which are equally important socio-economic status indicators, are measured by three categories based on the highest level of education completed. In this analysis, higher education, secondary education, and primary education binary variables are used, with an education level less than primary education serving as the benchmark category. Evidence on the inverse relationship between non-communicable diseases and education has been accumulated in developing countries [48–50].

In India, regional heterogeneity has been known as a strong contributor to inequalities in health and healthcare access [30]. To control for differences in cultural and economic conditions across states, this study includes binary state fixed effects. Delhi and the union territories serve as the benchmark for this analysis. Descriptive statistics of outcome variables and covariates used in this study are shown in Table 1.

## Measuring dissimilarity

This study quantifies socioeconomic-related inequality in binary outcome variables using the dissimilarity index (D-index) [51]. The D-index compares the mean outcomes of different

**Table 1. Descriptive statistics of outcomes and covariates.**

| | (1) | | | (2) | | |
|---|---|---|---|---|---|---|
| | Pre-pandemic period | | | Pandemic period | | |
| | count | mean | sd | count | mean | sd |
| Hypertension | 80849 | 0.30 | 0.46 | 486376 | 0.29 | 0.45 |
| Diabetes | 78002 | 0.05 | 0.21 | 471024 | 0.04 | 0.21 |
| Medication against hypertension | 24411 | 0.22 | 0.42 | 139423 | 0.18 | 0.38 |
| Medication against diabetes | 3517 | 0.69 | 0.46 | 20935 | 0.62 | 0.49 |
| Male | 96744 | 0.50 | 0.50 | 565815 | 0.49 | 0.50 |
| Age 30–39 | 96744 | 0.23 | 0.42 | 565815 | 0.22 | 0.42 |
| Age 40–49 | 96744 | 0.18 | 0.38 | 565815 | 0.19 | 0.39 |
| Age 50–59 | 96744 | 0.14 | 0.34 | 565815 | 0.15 | 0.36 |
| Age over 60 | 96744 | 0.18 | 0.38 | 565815 | 0.18 | 0.38 |
| Scheduled caste | 96744 | 0.24 | 0.43 | 565815 | 0.21 | 0.41 |
| Scheduled tribe | 96744 | 0.09 | 0.29 | 565815 | 0.18 | 0.39 |
| Other backward class | 96744 | 0.39 | 0.49 | 565815 | 0.40 | 0.49 |
| Wealth 2nd quintile | 96744 | 0.20 | 0.40 | 565815 | 0.19 | 0.40 |
| Wealth 3rd quintile | 96744 | 0.20 | 0.40 | 565815 | 0.18 | 0.38 |
| Wealth 4th quintile | 96744 | 0.21 | 0.41 | 565815 | 0.17 | 0.38 |
| Wealth 5th quintile | 96744 | 0.24 | 0.43 | 565815 | 0.22 | 0.42 |
| Primary education | 96744 | 0.14 | 0.35 | 565815 | 0.14 | 0.35 |
| Secondary education | 96744 | 0.39 | 0.49 | 565815 | 0.40 | 0.49 |
| Higher education | 96744 | 0.16 | 0.37 | 565815 | 0.15 | 0.36 |
| Observations | 96744 | | | 565815 | | |

Pre-pandemic period: June 17, 2019—January 30, 2020

Pandemic period: October 12, 2020—May 20, 2021

socio-economic groups with the mean outcome for the entire population, measuring the extent of dissimilarity in outcomes based on SES. The D-index is calculated as follows: when $Y = \{0, 1\}$ denotes the binary outcome,

$$D(\widehat{p}_i) = \frac{1}{2\overline{p}} \sum_{i=1}^{n} \frac{1}{n} |\widehat{p}_i - \overline{p}|, \tag{2}$$

where $n$ is the sample size; $\widehat{p}_i$ is the predicted probability of $Y = 1$ for individual $i$ by the multivariate logistic regression; and $\overline{p} = \frac{1}{n}\sum_{i=1}^{n} \widehat{p}_i$. The latter part of Eq (2), $\sum_{i=1}^{n} \frac{1}{n} |\widehat{p}_i - \overline{p}|$, represents the average absolute disparity of individual predicted probabilities from the average. By dividing this average absolute disparity by $2\overline{p}$, the D-index indicates the relative degree of dissimilarity, ranging from 0 to 1, with 0 indicating an equal chance of having $Y = 1$ across different groups. Therefore, the D-index measures the relative proportion of opportunities that need to be redistributed from better-off to worse-off groups (or vice versa) to achieve a uniform distribution of predicted probabilities.

## Shapley decomposition of the D-index

Understanding the sources of inequality is crucial for policymakers aiming to identify their determinants and explore potential governmental interventions to mitigate them. The D-index can be broken down into proportions attributable to various SES factors using the

Shapley decomposition method [52]. This method calculates the expected marginal contributions of each independent variable, determining the marginal impacts of each variable on the overall inequality of the outcome variable.

The Shapley value decomposition of the D-index operates as follows. First, select a variable, say $X_1$, and replace its original values with its sample mean, $\overline{X_1}$, for all individuals. Using $\overline{X_1}$, predict the hypothetical probability for each person with the estimated probability model, $\widehat{p}_{-1i} = \widehat{f}(\overline{X_1}, X_{2i}, \ldots, X_{qi})$, from which $D(\widehat{p}_{-1i})$ is calculated. The marginal contribution of $X_1$ to total inequality is then qualified by the difference between the D-index calculated with the original predicted probability and that with the hypothetical predicted probability, $D(\widehat{p}_i) - D(\widehat{p}_{-1i})$. This approach, intuitively, estimates the contribution of $X_1$ by subtracting the hypothetical inequality level in which the effect of $X_1$ is suppressed, from the overall observed inequality. In the following step, replace another variable, say $X_2$, and calculate its marginal contribution similarly. Namely, this step calculates $D(\widehat{p}_{-1i}) - D(\widehat{p}_{-1-2i})$, where $\widehat{p}_{-1-2i} = \widehat{f}(\overline{X_1}, \overline{X_2}, X_{3i}, \ldots, X_{qi})$. Continue this process for all independent variables, replacing each with its sample mean, to determine how much inequality is marginally reduced when nullifying the effect of each circumstance variable.

However, the magnitude of each marginal contribution can vary depending on the order in which it is calculated. Considering all possible orders results in numerous marginal contributions for each variable. By averaging all these possible marginal contributions, this approach obtains the expected marginal contribution of each variable, known as the Shapley value [52]. The advantage of Shapley value decomposition is that the order in which each covariate is nullified does not influence the final result. Additionally, the sum of all contributions equals the overall D-index value.

## Ethical statement

The International Institute for Population Sciences (IIPS), Mumbai, provided the ethical approval of NFHS-5 (2019–21). Additionally, the ICF International Review Board (IRB) looked over the survey and gave ethical approval. The respondents provided signed consent after being fully informed about the survey's purpose and procedures. Only interviews were done after obtaining proper consent from each participant. The Demographic and Health Surveys (DHS) Program's website hosts the NFHS-5, an anonymous dataset that is made available to the public and cannot be used to identify the survey respondents.

## Results

### Regression analysis

**Hypertension prevalence and medication.** Table 2 presents the multivariate logistic regression results concerning hypertension prevalence and medication, before the COVID-19 pandemic and during the pandemic respectively. The estimated odds ratios, along with their 95% confidence intervals, are reported. Regarding hypertension prevalence, significant associations are found with gender, age, wealth and education. Age emerges as the most influential risk factor, with older individuals exhibiting a higher prevalence. Notably, those in the highest quintile of household wealth face 1.53 times higher risk compared to the lowest quintile during the pandemic. With respect to educational gradient, higher education shows a lower risk than no educational qualification both before and during the pandemic. Disadvantaged tribes show significant associations with higher risk, albeit its association much smaller compared to other demographic and socio-economic factors.

**Table 2. Regression analysis for hypertension prevalence and its treatment.**

| | (1) | | (2) | | (3) | | (4) | |
|---|---|---|---|---|---|---|---|---|
| | Hypertension prevalence pre-pandemic period | | Hypertension prevalence pandemic period | | Hypertension medication pre-pandemic period | | Hypertension medication pandemic period | |
| Odds ratio | | | | | | | | |
| Male | 1.321*** | [1.277,1.367] | 1.275*** | [1.257,1.294] | 0.448*** | [0.417,0.481] | 0.501*** | [0.485,0.517] |
| Age 30–39 | 1.966*** | [1.863,2.075] | 2.168*** | [2.116,2.222] | 0.851** | [0.736,0.983] | 0.885*** | [0.816,0.960] |
| Age 40–49 | 3.290*** | [3.114,3.476] | 3.898*** | [3.805,3.994] | 1.296*** | [1.131,1.485] | 1.435*** | [1.332,1.545] |
| Age 50–59 | 4.883*** | [4.604,5.178] | 6.527*** | [6.364,6.695] | 1.935*** | [1.690,2.216] | 2.402*** | [2.235,2.582] |
| Age over 60 | 7.279*** | [6.874,7.708] | 9.647*** | [9.406,9.894] | 3.048*** | [2.676,3.471] | 3.924*** | [3.656,4.212] |
| Scheduled caste | 0.952** | [0.908,0.997] | 0.983 | [0.961,1.005] | 0.801*** | [0.731,0.878] | 0.862*** | [0.824,0.903] |
| Scheduled tribe | 0.904*** | [0.843,0.969] | 1.084*** | [1.057,1.111] | 0.676*** | [0.582,0.786] | 0.569*** | [0.535,0.605] |
| Other backward class | 0.840*** | [0.806,0.875] | 0.917*** | [0.899,0.935] | 0.797*** | [0.736,0.863] | 0.873*** | [0.840,0.909] |
| Wealth 2nd quintile | 1.126*** | [1.063,1.191] | 1.055*** | [1.033,1.078] | 1.232*** | [1.082,1.404] | 1.440*** | [1.361,1.525] |
| Wealth 3rd quintile | 1.266*** | [1.195,1.341] | 1.173*** | [1.146,1.200] | 1.445*** | [1.270,1.643] | 1.789*** | [1.690,1.894] |
| Wealth 4th quintile | 1.430*** | [1.347,1.518] | 1.328*** | [1.296,1.361] | 1.543*** | [1.355,1.758] | 2.070*** | [1.953,2.194] |
| Wealth 5th quintile | 1.711*** | [1.604,1.826] | 1.534*** | [1.493,1.577] | 2.077*** | [1.814,2.378] | 2.552*** | [2.399,2.715] |
| Primary education | 1.029 | [0.978,1.082] | 1.003 | [0.983,1.025] | 1.184*** | [1.073,1.306] | 1.366*** | [1.307,1.427] |
| Secondary education | 0.968 | [0.925,1.014] | 0.982* | [0.963,1.001] | 1.335*** | [1.218,1.464] | 1.422*** | [1.363,1.483] |
| Higher education | 0.857*** | [0.803,0.914] | 0.857*** | [0.833,0.881] | 1.417*** | [1.244,1.614] | 1.550*** | [1.458,1.649] |
| Urban | 1.108*** | [1.065,1.153] | 1.124*** | [1.104,1.144] | 1.079** | [1.000,1.164] | 1.188*** | [1.147,1.230] |
| Southern | 0.893*** | [0.824,0.967] | 1.061*** | [1.036,1.086] | 2.420*** | [2.099,2.789] | 2.143*** | [2.049,2.242] |
| Northern | 0.933*** | [0.899,0.969] | 1.030*** | [1.010,1.050] | 1.117*** | [1.036,1.205] | 1.118*** | [1.073,1.165] |
| North eastern | 1.461*** | [1.338,1.596] | 1.509*** | [1.463,1.557] | 1.640*** | [1.373,1.959] | 1.296*** | [1.199,1.401] |
| Eastern | 1.031 | [0.970,1.095] | 0.916*** | [0.898,0.934] | 1.607*** | [1.429,1.808] | 1.638*** | [1.566,1.715] |
| State fixed effect | Yes | | Yes | | Yes | | Yes | |
| Observations | 80849 | | 486376 | | 24411 | | 139423 | |

Exponentiated coefficients; 95% confidence intervals in brackets

Standard errors are robust to heteroskedasticity

Pre-pandemic period: June 17, 2019—January 30, 2020

Pandemic period: Obtober 12, 2020—May 20, 2021

*$p < 0.1$,

**$p < 0.05$,

***$p < 0.01$

Moving to the medication for hypertension, being female and older are associated with higher odds of receiving a proper medication. Age remains the strongest factor, indicating that a large proportion of young individuals with high blood pressure remain unmedicated. Substantial socio-economic gradients are observed, with wealthier and more educated individuals more likely to receive a proper treatment. Intriguingly these gradients are far steeper during the pandemic. For instance, while before the pandemic the odds ratio of higher education is 1.42, it becomes 1.55 amid the pandemic. Moreover, social class plays a significant role, with those from disadvantaged castes or tribes being less likely to receive a treatment.

**Diabetes prevalence and medication.** Table 3 presents the multivariate logistic regression results for diabetes prevalence and medication before the COVID-19 pandemic and during the pandemic respectively. For prevalence, strong associations are found with gender, age, wealth and education, revealing clearer and steeper socio-economic gradients compared to hypertension. During the pandemic, individuals in the richest quintile of household wealth are

**Table 3. Regression analysis for diabetes prevalence and its treatment.**

| | (1) | | (2) | | (3) | | (4) | |
|---|---|---|---|---|---|---|---|---|
| | Diabetes prevalence pre-pandemic period | | Diabetes prevalence pandemic period | | Diabetes medication pre-pandemic period | | Diabetes medication pandemic period | |
| Odds ratio | | | | | | | | |
| Male | 0.988 | [0.916,1.066] | 0.955*** | [0.926,0.985] | 0.782*** | [0.666,0.919] | 0.806*** | [0.756,0.859] |
| Age 30–39 | 2.815*** | [2.354,3.367] | 2.759*** | [2.546,2.990] | 0.493*** | [0.330,0.735] | 0.334*** | [0.281,0.397] |
| Age 40–49 | 6.199*** | [5.226,7.353] | 6.700*** | [6.212,7.226] | 0.545*** | [0.371,0.800] | 0.435*** | [0.370,0.512] |
| Age 50–59 | 11.79*** | [9.951,13.98] | 13.06*** | [12.12,14.08] | 0.761 | [0.518,1.118] | 0.634*** | [0.540,0.745] |
| Age over 60 | 15.44*** | [13.05,18.26] | 17.25*** | [16.01,18.58] | 0.930 | [0.637,1.360] | 0.884 | [0.753,1.038] |
| Scheduled caste | 0.835*** | [0.754,0.926] | 0.923*** | [0.882,0.967] | 0.817* | [0.661,1.009] | 0.881*** | [0.803,0.966] |
| Scheduled tribe | 0.569*** | [0.469,0.689] | 0.583*** | [0.546,0.623] | 0.611** | [0.416,0.900] | 0.772*** | [0.677,0.881] |
| Other backward class | 0.779*** | [0.714,0.850] | 0.886*** | [0.851,0.921] | 0.927 | [0.771,1.114] | 0.918** | [0.847,0.994] |
| Wealth 2nd quintile | 1.362*** | [1.163,1.595] | 1.312*** | [1.237,1.391] | 0.968 | [0.698,1.341] | 1.177*** | [1.046,1.324] |
| Wealth 3rd quintile | 1.572*** | [1.345,1.837] | 1.714*** | [1.617,1.817] | 0.967 | [0.702,1.333] | 1.376*** | [1.225,1.546] |
| Wealth 4th quintile | 2.025*** | [1.735,2.363] | 2.113*** | [1.992,2.241] | 0.934 | [0.679,1.284] | 1.455*** | [1.294,1.636] |
| Wealth 5th quintile | 2.601*** | [2.215,3.055] | 2.734*** | [2.569,2.910] | 1.111 | [0.798,1.549] | 1.943*** | [1.715,2.202] |
| Primary education | 1.232*** | [1.102,1.378] | 1.224*** | [1.171,1.279] | 1.381*** | [1.093,1.745] | 1.214*** | [1.110,1.328] |
| Secondary education | 1.464*** | [1.324,1.618] | 1.374*** | [1.318,1.431] | 1.312*** | [1.067,1.614] | 1.298*** | [1.194,1.410] |
| Higher education | 1.378*** | [1.198,1.584] | 1.250*** | [1.177,1.326] | 1.678*** | [1.253,2.246] | 1.343*** | [1.191,1.515] |
| Urban | 1.369*** | [1.260,1.486] | 1.226*** | [1.185,1.269] | 1.090 | [0.919,1.293] | 1.257*** | [1.174,1.347] |
| Southern | 3.346*** | [2.963,3.777] | 2.606*** | [2.503,2.714] | 1.700*** | [1.320,2.191] | 2.046*** | [1.882,2.224] |
| Northern | 0.834*** | [0.765,0.909] | 0.992 | [0.952,1.035] | 1.102 | [0.920,1.320] | 1.240*** | [1.138,1.353] |
| North eastern | 1.181 | [0.940,1.485] | 1.100** | [1.003,1.206] | 1.053 | [0.662,1.674] | 0.948 | [0.789,1.141] |
| Eastern | 1.711*** | [1.513,1.935] | 1.370*** | [1.309,1.434] | 1.132 | [0.879,1.457] | 1.069 | [0.977,1.170] |
| State fixed effect | Yes | | Yes | | Yes | | Yes | |
| Observations | 78002 | | 471024 | | 3517 | | 20935 | |

Exponentiated coefficients; 95% confidence intervals in brackets

Standard errors are robust to heteroskedasticity

Pre-pandemic period: June 17, 2019—January 30, 2020

Pandemic period: Obtober 12, 2020—May 20, 2021

*$p < 0.1$,

**$p < 0.05$,

***$p < 0.01$

2.73 times more likely to have high blood glucose levels compared to those in the poorest quintile. Similarly, those with higher education have a 1.25 times higher risk compared to individuals with less than a primary education during the pandemic. Wealth-related and education-related gradients, however, did not change very much during the pandemic. A gradient is also observed in terms of social class, with individuals from disadvantaged backgrounds having a lower risk.

Concerning medication, pro-rich wealth gradients, where individuals from more advantaged socio-economic backgrounds more likely to control their diabetes with medication, become apparent during the pandemic. For instance, the odds ratio for individuals in the richest quintile relative to the poorest quintile rises from 1.11 to 1.94. Additionally, individuals from disadvantaged social classes have a lower probability of taking medication to control elevated blood glucose levels. On the other hand, the deterioration of education-related inequality were not observed during the pandemic.

Overall, the regression analysis highlights robust evidence of socio-economic gradients in the prevalence and treatment of hypertension and diabetes in both periods. Particularly steeper pro-rich gradients are evident in terms their treatment, with older women from higher socio-economic backgrounds being more likely to control hypertension and diabetes with prescribed medication. Wealth and educational gradients in the treatment are more evident during the pandemic, suggesting the deterioration of socio-economic gradients amid the pandemic. One notable difference between hypertension and diabetes is that being older is not associated with a higher probability of managing diabetes. In fact, younger individuals under the age of 30 with diabetes are more likely to treat their condition. One plausible explanation could be that these younger individuals may suffer from congenital type-I diabetes, which is typically diagnosed earlier in life, rather than acquired type-II diabetes. As individuals age, acquired type-II diabetes becomes more common, potentially resulting in a smaller proportion of individuals with elevated blood glucose levels treating their condition as type-II diabetes may not exhibit symptoms until later stages. Another intriguing finding concerns gender differences. Although men are more likely to have high blood pressure, they are less likely to receive treatment.

## Socio-economic inequality and decomposition analysis of dissimilarity index

Fig 1 illustrates the estimated D-index values for the prevalence and treatment of hypertension and diabetes across age groups: below 30, 30–39, 40–49, 50–59, and 60 and over. Fig 1 shows that inequality in prevalence is relatively stable over time, but it shows notable increases in inequality of medication treatment for hypertension and diabetes among the middle-age group during the pandemic. In particular, among those aged between 50 and 59 and aged over 60, their estimated D-index values indicate significant increase, suggesting the deterioration of their socio-economic inequality in medication during the pandemic. The following subsections focus on the decomposition results of the D-index during the pandemic and discuss the contributory factors to the observed inequalities.

**Hypertension prevalence and medication.** Table 4 presents estimated D-index values and decomposition results regarding hypertension prevalence and treatment across different age groups. For hypertension prevalence, individuals below 30 show the highest D-index at 0.165, with a decline in D-index as age increases, indicating that hypertension becomes more common irrespective of socio-economic status as people age. Gender contributes to variation in prevalence probability, particularly notable among those below 30, but its contribution tends to diminish with age. Among the young people aged below 30, the relative contribution size made by gender is 78.5%, while it diminished to 9.6% for those aged over 60 years. Conversely, the contributions of education and wealth gradients become more prominent with age, suggesting increasing socioeconomic inequality as people age. Location of residence is also a significant contributor, especially for middle-aged and older individuals. It becomes the largest contributing factor, accounting for around 40% the total D-index among those in their 40s and 50s.

In terms of medication for hypertension, the contribution of each factor varies significantly across age groups. For those under 30, a large portion of dissimilarity is due to gender differences. However, as individuals age, the relative contribution of gender diminishes, while the impact of wealth and education gradients becomes more and more pronounced. Among those under 30, the relative contribution of gender is 87.4%, wealth is 3.2% and education is 2.7%. In contrast, for those over 60, the relative contributions shift to 10.3% for gender, 31.1% for wealth and 13.3% for education. The relative contribution made by location of residence also become greater as individuals age.

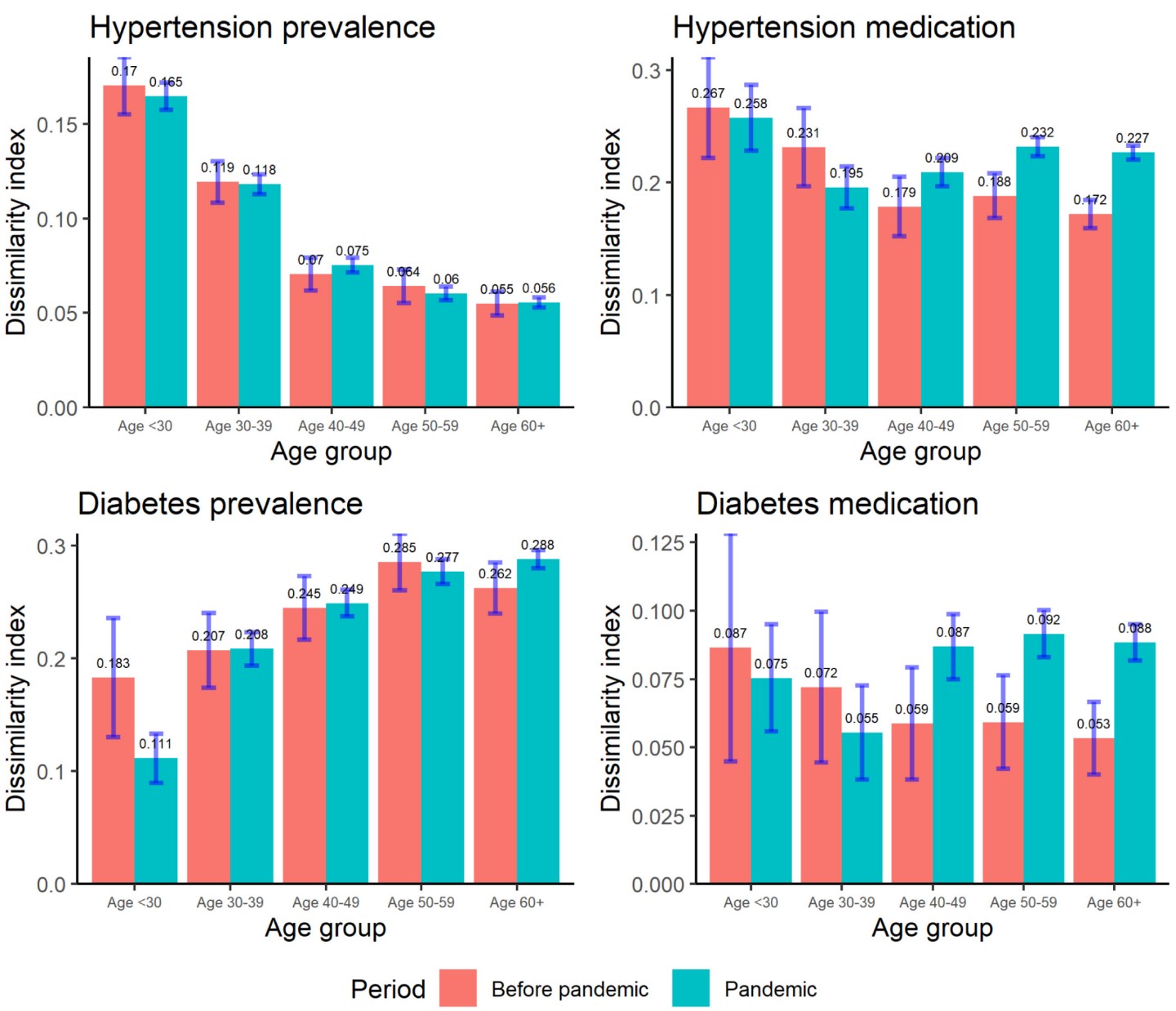

**Fig 1. The D-index values across age groups before and during the pandemic period.** Note: Bars indicate the 95% confidence intervals calculated by bootstrap with 200 repetitions. Pre-pandemic period: June 17, 2019—January 30, 2020. Pandemic period: October 12, 2020—May 20, 2021.

**Diabetes prevalence and medication.** Table 5 presents D-index values and decomposition results for diabetes prevalence and its treatment across age groups. For diabetes prevalence, the dissimilarity index values become larger for older people. Individuals aged over 60 exhibit the highest D-index value at 0.288. A steeper wealth gradient is observed among older individuals, with the wealth-related variation in prevalence increasing from 24.6% among those below 30 to over 30.0% among middle-aged individuals. Similarly, the education gradient shows a similar pattern, with its relative contribution increasing with age. The relative contribution made by education is 6.6%, 13.2%, 15.7%, 16.7% and 20.2% for the respective age groups. Conversely, the relative contributions of gender and social class decrease with age.

**Table 4. Dissimilarity index for hypertension across age groups during the pandemic.**

| Hypertension | Hypertension prevalence | | | Hypertension medication | | |
|---|---|---|---|---|---|---|
| | Estimates | 95 percent CIs | Proportion | Estimates | 95 percent CIs | Proportion |
| **Age below 30** | | | | | | |
| Dissimilarity index | 0.165*** | (0.157,0.172) | 1.000 | 0.258*** | (0.228,0.287) | 1.000 |
| Gender | 0.129*** | (0.121,0.137) | 0.785 | 0.225*** | (0.195,0.255) | 0.874 |
| Caste and tribe | 0.008*** | (0.005,0.011) | 0.050 | 0.005* | (-0.001,0.011) | 0.019 |
| Wealth | 0.008*** | (0.005,0.012) | 0.051 | 0.008** | (0.001,0.015) | 0.032 |
| Education | 0.003*** | (0.001,0.005) | 0.017 | 0.007*** | (0,0.014) | 0.027 |
| Location of living | 0.016*** | (0.011,0.02) | 0.096 | 0.012*** | (0.004,0.021) | 0.047 |
| **Age 30–39** | | | | | | |
| Dissimilarity index | 0.118*** | (0.113,0.123) | 1.000 | 0.195*** | (0.177,0.214) | 1.000 |
| Gender | 0.073*** | (0.067,0.078) | 0.615 | 0.119*** | (0.098,0.141) | 0.610 |
| Caste and tribe | 0.008*** | (0.006,0.01) | 0.066 | 0.019*** | (0.01,0.028) | 0.097 |
| Wealth | 0.013*** | (0.01,0.017) | 0.114 | 0.027*** | (0.017,0.037) | 0.137 |
| Education | 0.006*** | (0.005,0.008) | 0.054 | 0.008*** | (0.002,0.014) | 0.041 |
| Location of living | 0.018*** | (0.014,0.021) | 0.152 | 0.023*** | (0.013,0.032) | 0.115 |
| **Age 40–49** | | | | | | |
| Dissimilarity index | 0.075*** | (0.072,0.079) | 1.000 | 0.209*** | (0.197,0.222) | 1.000 |
| Gender | 0.02*** | (0.017,0.023) | 0.263 | 0.065*** | (0.055,0.076) | 0.313 |
| Caste and tribe | 0.007*** | (0.006,0.009) | 0.096 | 0.022*** | (0.016,0.028) | 0.107 |
| Wealth | 0.02*** | (0.014,0.026) | 0.268 | 0.056*** | (0.041,0.07) | 0.266 |
| Education | 0.01*** | (0.008,0.013) | 0.133 | 0.015*** | (0.01,0.02) | 0.072 |
| Location of living | 0.018*** | (0.014,0.022) | 0.240 | 0.051*** | (0.038,0.063) | 0.242 |
| **Age 50–59** | | | | | | |
| Dissimilarity index | 0.06*** | (0.057,0.064) | 1.000 | 0.232*** | (0.223,0.24) | 1.000 |
| Gender | 0.001*** | (0,0.002) | 0.019 | 0.035*** | (0.029,0.04) | 0.150 |
| Caste and tribe | 0.004*** | (0.003,0.005) | 0.069 | 0.039*** | (0.035,0.042) | 0.167 |
| Wealth | 0.022*** | (0.018,0.025) | 0.358 | 0.068*** | (0.062,0.074) | 0.293 |
| Education | 0.008*** | (0.007,0.009) | 0.134 | 0.029*** | (0.023,0.035) | 0.124 |
| Location of living | 0.025*** | (0.023,0.028) | 0.420 | 0.062*** | (0.052,0.071) | 0.266 |
| **Age 60+** | | | | | | |
| Dissimilarity index | 0.056*** | (0.053,0.058) | 1.000 | 0.227*** | (0.22,0.233) | 1.000 |
| Gender | 0.005*** | (0.004,0.007) | 0.096 | 0.023*** | (0.02,0.026) | 0.103 |
| Caste and tribe | 0.004*** | (0.004,0.005) | 0.080 | 0.033*** | (0.027,0.039) | 0.146 |
| Wealth | 0.021*** | (0.019,0.024) | 0.383 | 0.071*** | (0.058,0.083) | 0.311 |
| Education | 0.005*** | (0.004,0.006) | 0.094 | 0.03*** | (0.023,0.037) | 0.133 |
| Location of living | 0.019*** | (0.017,0.022) | 0.346 | 0.07*** | (0.054,0.086) | 0.308 |

Note: 95 percent confidence intervals (CIs) are calculated by bootstrap with 200 repetitions. Contributory categories are defined as follows: 1) Gender: male (binary variable), 2) Caste and Tribe: scheduled caste, scheduled tribe, and other backward class, 3) Wealth: wealth quintiles, 4) Education: primary, secondary, and higher educational achievement, 5) Location of living: urban and regional (binary variables). Pre-pandemic period: June 17, 2019—January 30, 2020. Pandemic period: October 12, 2020—May 20, 2021.

*$p < 0.1$,

**$p < 0.05$,

***$p < 0.01$

**Table 5. Dissimilarity index for diabetes across age groups during the pandemic.**

| Diabetes | Diabetes prevalence | | | Diabetes medication | | |
|---|---|---|---|---|---|---|
| | Estimates | 95 percent CIs | Proportion | Estimates | 95 percent CIs | Proportion |
| **Age below 30** | | | | | | |
| Dissimilarity index | 0.111*** | (0.09,0.133) | 1.000 | 0.075*** | (0.056,0.095) | 1.000 |
| Gender | 0.003 | (-0.005,0.012) | 0.031 | 0.003 | (-0.005,0.011) | 0.043 |
| Caste and tribe | 0.019*** | (0.006,0.031) | 0.168 | 0.005 | (-0.003,0.013) | 0.063 |
| Wealth | 0.027*** | (0.012,0.043) | 0.246 | 0.01** | (0,0.02) | 0.136 |
| Education | 0.007 | (-0.003,0.017) | 0.066 | 0.014*** | (0.004,0.025) | 0.188 |
| Location of living | 0.054*** | (0.034,0.075) | 0.489 | 0.043*** | (0.022,0.064) | 0.570 |
| **Age 30–39** | | | | | | |
| Dissimilarity index | 0.208*** | (0.194,0.223) | 1.000 | 0.055*** | (0.038,0.073) | 1.000 |
| Gender | 0.012*** | (0.005,0.018) | 0.056 | 0.014*** | (0.002,0.027) | 0.257 |
| Caste and tribe | 0.032*** | (0.025,0.039) | 0.154 | 0.006 | (-0.003,0.015) | 0.112 |
| Wealth | 0.067*** | (0.051,0.083) | 0.324 | 0.012** | (0.001,0.024) | 0.225 |
| Education | 0.028*** | (0.018,0.037) | 0.132 | 0.008 | (-0.001,0.016) | 0.137 |
| Location of living | 0.07*** | (0.053,0.086) | 0.334 | 0.015** | (0,0.03) | 0.269 |
| **Age 40–49** | | | | | | |
| Dissimilarity index | 0.249*** | (0.237,0.261) | 1.000 | 0.087*** | (0.075,0.099) | 1.000 |
| Gender | 0.004** | (0.001,0.007) | 0.016 | 0.004** | (-0.001,0.009) | 0.048 |
| Caste and tribe | 0.036*** | (0.032,0.04) | 0.146 | 0.007*** | (0.004,0.009) | 0.075 |
| Wealth | 0.085*** | (0.068,0.101) | 0.340 | 0.028*** | (0.021,0.036) | 0.324 |
| Education | 0.039*** | (0.023,0.055) | 0.157 | 0.012*** | (0.006,0.018) | 0.141 |
| Location of living | 0.085*** | (0.055,0.115) | 0.342 | 0.036*** | (0.027,0.045) | 0.412 |
| **Age 50–59** | | | | | | |
| Dissimilarity index | 0.277*** | (0.266,0.288) | 1.000 | 0.092*** | (0.083,0.1) | 1.000 |
| Gender | 0.003*** | (0.001,0.006) | 0.012 | 0.008*** | (0.004,0.011) | 0.082 |
| Caste and tribe | 0.04*** | (0.029,0.051) | 0.143 | 0.007*** | (0.004,0.01) | 0.079 |
| Wealth | 0.092*** | (0.067,0.116) | 0.330 | 0.023*** | (0.018,0.028) | 0.250 |
| Education | 0.046*** | (0.033,0.06) | 0.167 | 0.011*** | (0.007,0.014) | 0.116 |
| Location of living | 0.096*** | (0.071,0.121) | 0.347 | 0.043*** | (0.033,0.054) | 0.473 |
| **Age 60+** | | | | | | |
| Dissimilarity index | 0.288*** | (0.28,0.296) | 1.000 | 0.088*** | (0.082,0.095) | 1.000 |
| Gender | 0.001 | (0,0.002) | 0.003 | 0.002** | (0,0.003) | 0.017 |
| Caste and tribe | 0.037*** | (0.034,0.04) | 0.129 | 0.01*** | (0.008,0.013) | 0.116 |
| Wealth | 0.09*** | (0.084,0.096) | 0.312 | 0.026*** | (0.021,0.03) | 0.290 |
| Education | 0.058*** | (0.054,0.063) | 0.202 | 0.016*** | (0.012,0.019) | 0.177 |
| Location of living | 0.102*** | (0.095,0.108) | 0.353 | 0.035*** | (0.03,0.041) | 0.400 |

Note: 95 percent confidence intervals (CIs) are calculated by bootstrap with 200 repetitions. Contributory categories are defined as follows: 1) Gender: male (binary variable), 2) Caste and Tribe: scheduled caste, scheduled tribe, and other backward class, 3) Wealth: wealth quintiles, 4) Education: primary, secondary, and higher educational achievement, 5) Location of living: urban and regional (binary variables). Pre-pandemic period: June 17, 2019—January 30, 2020. Pandemic period: October 12, 2020—May 20, 2021.

*$p < 0.1$,

**$p < 0.05$,

***$p < 0.01$

For proper medication against diabetes, the D-index values are smaller than those observed for hypertension medication, suggesting that the dissimilarity in the probability of taking pre-scribed medicine for elevated blood glucose is less influenced by individual demographic and socio-economic characteristics. In other words, the low dissimilarity index and high average value of proper medication imply that people generally adhere to diabetic medication regardless of their individual characteristics. A significant portion of the D-index is attributed to location differences. The relative contribution of location decreases with age. Among those under 30, 57.0% of the D-index is associated with location, whereas this decreases to 40.0% among those over 60 years old. Wealth inequality also plays a role in diabetic medication, with a greater contribution being observed among middle-aged and older individuals, mirroring patterns seen in hypertension medication. Interestingly, social class and educational attainment do not heavily contribute to the variation in diabetic medication across age groups, with their relative contributions around 10%.

The corresponding results regarding the prevalence and treatment of hypertension and diabetes prior to the pandemic are presented in S1 Appendix. Additional results for the pre-pandemic period, based on the previous wave of the NFHS conducted in 2015–2016 (NFHS-4) are provided in the S1 Appendix for reference. The results are largely consistent with those observed during the pandemic period. As shown in Fig 1, socioeconomic disparities in treatment are notably more pronounced among middle-aged groups during the pandemic. A comparison with decomposition results from both periods reveals that the relative contribution of wealth to these disparities is higher during the pandemic, particularly for diabetes medication. This finding suggests that the exacerbation of socioeconomic inequality in diabetes medication is strongly linked to reduced access to treatment among less wealthy individuals.

## Discussion

With economic growth and its consequential lifestyle changes, India is facing an increase in the prevalence of NCDs [29]. However, most of the people in the country do not control their hypertension and diabetes [53], which could have been exacerbated during the COVID-19 pandemic [16]. The estimates showed that during the pandemic fewer than one-fifth of people with high blood pressure are controlling hypertension and fewer than two-thirds of people with high glucose levels are controlling diabetes. Effective treatment at an earlier stage is crucial for minimising the impact of hypertension and diabetes on health and well-being as inadequately treated hypertension and diabetes for a long time could lead to worse health outcomes. Uncontrolled hypertension and diabetes could incur significant economic costs for patients, families, health systems and national economies.

India has tried to address the growing burden of NCDs. One of the ongoing comprehensive initiatives is the National Programme for Prevention and Control of Cancer, Diabetes, Cardiovascular Diseases, and Stroke (NPCDCS), launched by the Indian government in 2010 across all states [54]. The programme prioritises enhancing healthcare infrastructure, building human resource capacity, promoting health awareness, and implementing population-based screening for individuals aged 30 and above. It emphasises early detection, effective management, and appropriate referral across various levels of healthcare. The programme's goal is to ensure that healthcare services for NCDs are accessible across all levels, from sub-centres to tertiary care facilities. For activities up to district level and below under NPCDCS, states are given financial support. However, despite its comprehensive framework, several issues have hindered its success in managing NCDs [55–57]. Primary health centres often lack essential resources like laboratory facilities and medications, while financial and managerial support is inadequate. As a result, the quality of care is

compromised, leading to poor compliance and discontinuation of treatment by patients. Moreover the programme faced significant challenges during the COVID-19 waves, leading to the suspension of non-COVID-19 medical services [38, 39]. Due to these shortfalls, a large proportion of patients would have continued to depend on private healthcare services for their NCDs management, which are often costly and inaccessible to many, potentially undermining the NPCDCS's goal of providing universal, affordable healthcare. This research focuses on the socio-economic gradients in management of hypertension and diabetes amid the Covid-19 pandemic and produced several new insights. From the perspective of socio-economic inequality, this study shows that hypertension and diabetes are more prevalent among people enjoying a more privileged socio-economic status. With respect to their management, the estimates showed that proper medication for hypertensive and diabetic conditions is more common among wealthy groups even after controlling for the respective conditions. Notable deterioration of socio-economic inequalities is observed among the middle-aged group during the pandemic. These results highlight that while wealthy and educated individuals have higher risks of suffering from these conditions, they are more likely to cope with them once they suffer from them. One straightforward interpretation is that wealthy people have superior access to healthcare and educated people have improved knowledge about the risks of NCDs and the importance of treating them. The higher risks that poorer households face with uncontrolled hypertension and diabetes imply they would be more likely to face future risks of life-threatening complications, which could lead to higher mortality or catastrophic health expenditures. In India, social security systems often do not cover all citizens or reimburse all essential medicines and medical devices. Consequently, most treatment costs, including medications, are paid out-of-pocket by individuals [58]. Lower SES groups are thus likely to be confronted with an enormous risk of facing extremely high medical expenditures, which further impoverishes them and keeps them in poverty.

The decomposition analysis shows that wealth- and education-related gradients become steeper for older age groups, thus suggesting that socio-economic inequalities in terms of medication are exacerbated as people become older. The contribution associated with the difference in individual educational attainments is not merely persistent even at middle ages but strengthens. On the other hand, the contributions made by gender indicate that they diminish as people age. For the case of hypertension medication, for instance, at a younger age, around 80% of the observed variation in the probabilities is associated with gender, but the relative contributory size rapidly diminishes in middle-aged groups and they ultimately become around 10% for people aged 60 and over. At the time people turn 50 years old, the contributions made by socio-economic status are greater than that of gender. Moreover, a comparison of the regression and decomposition results before and during the pandemic indicates that the worsening socioeconomic inequality in medication was closely associated with an increased wealth gradient. This underscores the greater importance of providing social support to less wealthy households requiring treatment to achieve horizontal equity in healthcare access during emergencies.

Limited effectiveness of the NPCDCS and these findings highlight that policymakers in India need to urgently consider policies to effectively improve healthcare access for all people, regardless of their demographic and socio-economic status, for each age group. Ensuring that people at risk of NCDs are aware of these risks, receive necessary diagnoses, and can manage their conditions will be essential for the government to address inequalities in unmet healthcare needs. These measures can be reinforced by age-specific, population-based public health strategies to enhance understanding of the necessity of treatment, leading to improved well-being. In addition, strengthening primary care services, ensuring resource availability,

improving managerial oversight, and boosting the financial investment in healthcare are critical steps needed to reduce the burden of NCDs in the country.

While demand-side interventions, such as raising public awareness and improving health literacy, are crucial, bolstering supply-side measures is equally important to enhance accessibility. In India, there are significant regional disparities in healthcare access and the characteristics of health providers, such as procurement, clinical practices, supply-chain systems and medication availability [30]. Supply-side policies are expected to improve access to necessary medical care and reduce treatment inequalities. A successful example of such an intervention is the India Hypertension Control Initiative (IHCI), which adopted the WHO-HEARTS strategy in 2018 and has been implemented in nearly 40 member states. This initiative shows that hypertension can be effectively controlled through large-scale national programmes based in primary healthcare facilities and communities [59]. These supply-side policies would be promising in resource-constrained areas to mitigate the urban-rural gap and address disparities in healthcare resource availability across regions.

## Conclusion

To conclude, limitations of this study must be addressed. Firstly, this study is a descriptive analysis that highlights existing socio-economic inequalities, and as such, the results do not necessarily imply causal relationships. Although the onset of COVID-19 was an exogenous shock, the decomposition results may be affected by unobservable confounding factors. Furthermore, since the observed inequality reflects the period when the Delta variant was prevalent and the pandemic had not yet stabilised, the results do not necessarily capture the ultimate inequality effects of the pandemic. However, understanding the current inequality is a crucial first step toward effectively designing necessary public policies to mitigate disparities in health and healthcare use.

Secondly, the dissimilarity index reflects socioeconomic inequality at the national level and does not capture inequality at the state or district level, making it impossible to compare inter-state or inter-district socioeconomic inequality. Additionally, the estimated inequality is based only on the states surveyed both before and during the pandemic. Although calculating the index for each state or district would technically be possible, this study could not pursue this approach due to limited sample sizes in the pandemic period after sample splitting. Investigating differences in inequality across states and exploring the factors contributing to inter-state inequality will be an agenda for future research.

Thirdly, although this study explores socio-economic inequality in health and healthcare access before and during the COVID-19 pandemic, the evidence presented is based on repeated cross-sectional analysis. This means the socio-economic disparities observed here represent snapshots of persistent inequality at two points in time. The relationships among socio-economic status, lifestyles, and health likely develop and reinforce each other over time. While significant pro-rich socio-economic gradients are observed for medication among younger individuals, the associated risks, such as the onset of complications and high medical costs, may not become apparent until middle or old age. Future research should investigate the long-term progression of these inequalities and their potential impacts on medical costs using longitudinal data. Studies should also consider socio-economic status throughout an individual's life and the lifelong accumulation of risk factors to fully understand their dynamic associations. If large-scale longitudinal data become available in India, researchers could examine the long-term effects of under-treatment in younger people on future medical burdens and the consequences on well-being.

## Supporting information

**S1 Appendix. Supplementary analysis.**
(PDF)

## Author Contributions

**Conceptualization:** Toshiaki Aizawa.

**Formal analysis:** Toshiaki Aizawa.

**Validation:** Toshiaki Aizawa.

**Writing – original draft:** Toshiaki Aizawa.

**Writing – review & editing:** Toshiaki Aizawa.

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
