## [Decision Letter · Decision Letter 0]

18 Sep 2024

PONE-D-24-30611Socio-economic Gradients in Hypertension and Diabetes Management Amid the COVID-19 Pandemic in IndiaPLOS ONE Dear Dr. Aizawa,

Thank you for submitting your manuscript to PLOS ONE. After careful consideration, we feel that it has merit but does not fully meet PLOS ONE’s publication criteria as it currently stands. Therefore, we invite you to submit a revised version of the manuscript that addresses the points raised during the review process.

As indicated by both reviewers, the author should include a comprehensive review of the literature and explicitly highlight the contribution of this study in light of the literature, in addition to addressing the other reviewers' comments. 

We look forward to receiving your revised manuscript.

Kind regards,

Bijetri Bose

Academic Editor

PLOS ONE

“JSPS Grant-in-Aid for Scientific Research (#24K1634504)”

3. Please note that your Data Availability Statement is currently missing the repository name and/or the DOI/accession number of each dataset OR a direct link to access each database. If your manuscript is accepted for publication, you will be asked to provide these details on a very short timeline. We therefore suggest that you provide this information now, though we will not hold up the peer review process if you are unable.

Reviewers' comments:

Reviewer's Responses to Questions

**Comments to the Author**

1. Is the manuscript technically sound, and do the data support the conclusions?

Reviewer #1: Partly

Reviewer #2: Yes

2. Has the statistical analysis been performed appropriately and rigorously? 

Reviewer #1: Yes

Reviewer #2: I Don't Know

3. Have the authors made all data underlying the findings in their manuscript fully available?

Reviewer #1: Yes

Reviewer #2: Yes

4. Is the manuscript presented in an intelligible fashion and written in standard English?

Reviewer #1: Yes

Reviewer #2: Yes

5. Review Comments to the Author

Reviewer #1: The article is well-written and addresses a critical issue about access to medication for diabetes and hypertension before and during the pandemic. The study aims to understand the effect of the pandemic on the socioeconomic inequalities in access to medication. Although the study carries out an extensive empirical analysis, it does not reveal the effects of the pandemic. Also, the study needs to include a comprehensive literature survey. Detailed comments about both points are given below:

1. The study uses NFHS-5 dataset for India. This survey was conducted in two phases due to the pandemic-related disturbance in the initial survey schedule. As a result, the two phases cover different districts/states. So, the observed differences in the socioeconomic inequalities in access to medication may result from inherent differences in these districts and not due to the pandemic. The author can utilize NFHS-4 data to make the comparison using the difference-in-difference method.

2. A comprehensive review of existing studies is required as some crucial references are missing. For example,

Press, J. ∙ Manne-Goehler, J. ∙ Jaacks, L.M. ∙ et al. Hypertension screening, awareness, treatment, and control in India: a nationally representative cross-sectional study among individuals aged 15 to 49 years PLoS Med. ; 16 (2019), e1002801

Rao Guthi, Visweswara et al. 2024. Hypertension treatment cascade among men and women of reproductive age group in India: analysis of National Family Health Survey-5 (2019–2021) The Lancet Regional Health - Southeast Asia, Volume 23, 100271 (2024).

Maiti, S., Akhtar, S., Upadhyay, A.K. et al. Socioeconomic inequality in awareness, treatment and control of diabetes among adults in India: Evidence from National Family Health Survey of India (NFHS), 2019–2021. Sci Rep 13, 2971 (2023). https://doi.org/10.1038/s41598-023-29978-y

3. Lastly, the authors should bring out the contribution of this study in light of the literature.

Reviewer #2: This paper examines the socioeconomic disparities in hypertension and diabetes prevalence and treatment in India before and during the COVID-19 pandemic. Using two waves of a nationally representative populations survey, the authors employ a Shapley decomposition analysis to identify the drivers of disadvantage and undertreatment of these illnesses. It is a relevant to distill the relative impact of demographic characteristics on health status and the analysis is technically sound. However, in its current form, the analysis makes a limited contribution to what is already know about this topic.

Major recommendations to strengthen the manuscript:

1. This paper could benefit by situating the issues in the context of the Indian health system and health policies. More details about NCD policies, different care providers, and levels of care seeking could help motivate and specify the research question. Specifically, are outcomes ameliorating for some demographic subgroups compared to others? These details would help motivate the current research question. But in its current form, the novelty of the findings seem limited.

2. Relatedly, the discussion section could go much further in discussing the policy and reform implications of the findings. Specifically, what has been tried so far to address the growing burden of hypertension and diabetes, and how do these findings address their limited success thus far?

3. As the authors mention, there is an extensive literature documenting the demographic determinants of NCDs globally. Yet in its current form, the manuscript has little purchase to answer new questions about the persistence and growing prevalence of NCDs in India.

6. PLOS authors have the option to publish the peer review history of their article (what does this mean?). If published, this will include your full peer review and any attached files.

Reviewer #1: No

Reviewer #2: No

---

## [Author Response · Author response to Decision Letter 0]

4 Oct 2024

Reply to reviewers’ comments

> Reviewer #1: The article is well-written and addresses a critical issue about access to medication for diabetes and hypertension before and during the pandemic. The study aims to understand the effect of the pandemic on the socioeconomic inequalities in access to medication. Although the study carries out an extensive empirical analysis, it does not reveal the effects of the pandemic. Also, the study needs to include a comprehensive literature survey. Detailed comments about both points are given below:

(Reply)

Thank you very much for reviewing my paper and offering valuable insights and suggestions. Your feedback has greatly enhanced the quality of the manuscript. I have highlighted the key revisions in the updated manuscript, and my detailed responses to your comments are outlined below.

>1. The study uses NFHS-5 dataset for India. This survey was conducted in two phases due to the pandemic-related disturbance in the initial survey schedule. As a result, the two phases cover different districts/states. So, the observed differences in the socioeconomic inequalities in access to medication may result from inherent differences in these districts and not due to the pandemic. The author can utilize NFHS-4 data to make the comparison using the difference-in-difference method.

(Reply)

Thank you for your comment and suggestion. Since the western regions were not surveyed during the pandemic period, they are also excluded from the analysis for the pre-pandemic period to ensure full comparability. This exclusion mitigates potential bias stemming from non-comparability due to unobservable regional differences. 

As you suggested, NFHS-4 can be used to provide evidence for the pre-pandemic period. One advantage is that it allows for a larger sample size when estimating the D-index before the pandemic. However, in NFHS-4, only men aged 15–49 and women aged 15–54 were surveyed, and the survey did not inquire about diabetes medication. Therefore, the revised manuscript provides additional results only on hypertension and its medication in NFHS-4 solely for reference purposes in the Appendix. 

>2. A comprehensive review of existing studies is required as some crucial references are missing. For example,

Press, J. ∙ Manne-Goehler, J. ∙ Jaacks, L.M. ∙ et al. Hypertension screening, awareness, treatment, and control in India: a nationally representative cross-sectional study among individuals aged 15 to 49 years PLoS Med. ; 16 (2019), e1002801

Rao Guthi, Visweswara et al. 2024. Hypertension treatment cascade among men and women of reproductive age group in India: analysis of National Family Health Survey-5 (2019–2021) The Lancet Regional Health - Southeast Asia, Volume 23, 100271 (2024).

Maiti, S., Akhtar, S., Upadhyay, A.K. et al. Socioeconomic inequality in awareness, treatment and control of diabetes among adults in India: Evidence from National Family Health Survey of India (NFHS), 2019–2021. Sci Rep 13, 2971 (2023). https://doi.org/10.1038/s41598-023-29978-y

(Reply)

Thank you for letting me know the important literature. In the revised manuscript, they are included in the literature review. I’ve also substantially updated the literature review by adding a few more studies about the socioeconomic inequality in hypertension and diabetes treatment.

>3. Lastly, the authors should bring out the contribution of this study in light of the literature.

(Reply)

Thank you for your comments. In the revised draft, I’ve extended the discussion and explicitly clarifies the contributions of this research to the existing literature at the end of the Introduction section. Especially, I discussed why analysing inequalities in healthcare use during the pandemic is important. Furthermore, the extended implications added in the discussion section also highlights the contribution of this research.

>Reviewer #2: This paper examines the socioeconomic disparities in hypertension and diabetes prevalence and treatment in India before and during the COVID-19 pandemic. Using two waves of a nationally representative populations survey, the authors employ a Shapley decomposition analysis to identify the drivers of disadvantage and undertreatment of these illnesses. It is a relevant to distill the relative impact of demographic characteristics on health status and the analysis is technically sound. However, in its current form, the analysis makes a limited contribution to what is already know about this topic.

(Reply)

Thank you very much for reviewing my paper and offering valuable insights and suggestions. Your feedback has greatly enhanced the quality of our work. I have highlighted the key revisions in the updated manuscript, and my detailed responses to your comments are outlined below.

> 1. This paper could benefit by situating the issues in the context of the Indian health system and health policies. More details about NCD policies, different care providers, and levels of care seeking could help motivate and specify the research question. Specifically, are outcomes ameliorating for some demographic subgroups compared to others? These details would help motivate the current research question. But in its current form, the novelty of the findings seem limited.

(Reply)

Thank you for your valuable comments. In the revised draft, I have restructured the introduction and added a new section that provides information on the healthcare system in India and its challenges. I have also discussed why socioeconomic inequalities in healthcare access occur and why addressing them is essential. Additionally, I have updated the literature review and clarified the contribution of this study. I hope these revisions effectively motivate the current research question.

>2. Relatedly, the discussion section could go much further in discussing the policy and reform implications of the findings. Specifically, what has been tried so far to address the growing burden of hypertension and diabetes, and how do these findings address their limited success thus far?

(Reply)

In the revised manuscript, I have discussed the ongoing comprehensive policy called the National Programme for Prevention and Control of Cancer, Diabetes, Cardiovascular Diseases, and Stroke (NPCDCS), launched by the Indian government in 2010. While this policy aims to address the growing burden of non-communicable diseases (NCDs), recent studies highlight its limited success due to a shortage of both physical and human resources. I have emphasized the importance of overcoming these supply-side challenges to mitigate the worsening socioeconomic inequality in access to treatment. I hope the newly added paragraphs in the discussion section provide necessary direction toward ensuring universally accessible treatment for all, regardless of socioeconomic background.

>3. As the authors mention, there is an extensive literature documenting the demographic determinants of NCDs globally. Yet in its current form, the manuscript has little purchase to answer new questions about the persistence and growing prevalence of NCDs in India.

(Reply)

Thank you for your comment. In the revised draft, I have discussed the distinct roles of public and private healthcare facilities in India and why socioeconomic inequality in healthcare persists. By focusing on the exogenous COVID-19 shock, this study provides new evidence on the exacerbation of socioeconomic inequality. To clarify the contribution of this study, I have added a new subsection outlining the trend of healthcare use for NCDs during the pandemic period.

Additionally, I have explicitly highlighted the research contribution to the literature and emphasized the value of analysing the sources of socioeconomic inequality in healthcare access, particularly in the context of the COVID-19 pandemic.

---

## [Editor Report · Decision Letter 1]

22 Oct 2024

PONE-D-24-30611R1Socio-economic Gradients in Hypertension and Diabetes Management Amid the COVID-19 Pandemic in IndiaPLOS ONE

Dear Dr. Aizawa,

Thank you for addressing the reviewer's comments and re-submitting your manuscript to PLOS ONE. After careful consideration, we feel that it has merit but some issues need further attention. Therefore, we invite you to submit a revised version of the manuscript that addresses the points raised during the review process. As reviewer 1 pointed out, NFHS 5 was carried out in different districts/states in the 2 phases and any observed differences in the outcomes may not be attributable to the pandemic. Consequently, one must be careful about the conclusions on ‘how socioeconomic gradients shifted during the pandemic’ based on such comparisons [aim 2 of the study]. Excluding the western regions does not solve the problem because of 2 reasons. First, there are many other states that were not surveyed during the pandemic period but have not been dropped from your analysis. Second, given health falls under the jurisdiction of the India states (not the federal government), inter-state/district comparisons (using difference-in-difference methods or state/district fixed effects) are common in the literature. Even if you retain only those states that were surveyed in both phases, do inter-state comparisons in the Indian context make sense? For example, can we compare the outcomes in Haryana to those in Tamil Nadu? This must be explained or clearly discussed in the manuscript. There is also a need for greater clarity on the contributions of the study relative to the existing literature, especially based on how you decide to address the above concern. It would be great to understand what the value add of this study is over the Rao Guthi, Visweswara et al. 2024 and Maiti et. Al, 2023 papers. Is the difference between the studies in terms of the samples (all states in the cited paper vs. during pandemic in the current study)? Or does the current study have methodological advantages over the existing ones? Please submit your revised manuscript by Dec 06 2024 11:59PM. If you will need more time than this to complete your revisions, please reply to this message or contact the journal office at plosone@plos.org. Please include the following items when submitting your revised manuscript:A rebuttal letter that responds to each point raised by the academic editor and reviewer(s). You should upload this letter as a separate file labeled 'Response to Reviewers'.A marked-up copy of your manuscript that highlights changes made to the original version. You should upload this as a separate file labeled 'Revised Manuscript with Track Changes'.An unmarked version of your revised paper without tracked changes. You should upload this as a separate file labeled 'Manuscript'.

We look forward to receiving your revised manuscript.

Kind regards,

Bijetri Bose

Academic Editor

PLOS ONE

---

## [Author Response · Author response to Decision Letter 1]

6 Nov 2024

Response to editor’s comments:

>As reviewer 1 pointed out, NFHS 5 was carried out in different districts/states in the 2 phases and any observed differences in the outcomes may not be attributable to the pandemic. Consequently, one must be careful about the conclusions on ‘how socioeconomic gradients shifted during the pandemic’ based on such comparisons [aim 2 of the study]. 

(Reply)

Thank you for your valuable feedback. As you mentioned, the observed difference may not have been caused directly by the pandemic itself. In the revised manuscript, I have clarified that this difference does not necessarily reflect the causal effect of the pandemic but rather shows the association between socioeconomic inequality in treatment and the pandemic period.

I have also revised the description of the study's second aim to avoid any potential impression that this study estimates the causal impact of the pandemic on inequality.

> Second, the study explores the socioeconomic gradients and exploring the underlying drivers of the observed inequality during the pandemic. Excluding the western regions does not solve the problem because of 2 reasons. First, there are many other states that were not surveyed during the pandemic period but have not been dropped from your analysis. Second, given health falls under the jurisdiction of the India states (not the federal government), inter-state/district comparisons (using difference-in-difference methods or state/district fixed effects) are common in the literature. Even if you retain only those states that were surveyed in both phases, do inter-state comparisons in the Indian context make sense? For example, can we compare the outcomes in Haryana to those in Tamil Nadu? This must be explained or clearly discussed in the manuscript.

(Reply)

Thank you for your comments. In the previous draft, I excluded the western region, as it was the only region with states completely missing during the pandemic period. However, as you noted, this may not have been sufficient. In the revised draft, I have included only those states that were surveyed in both the pre-pandemic and pandemic periods to ensure comparability. The limited generalizability of the result due to this sample selection is acknowledged in the discussion. I re-ran the analysis and reinterpreted the results, which now show a worsening of socioeconomic inequalities in the treatment of hypertension and diabetes among middle-aged individuals.

To address potential differences across states, my analysis controls for state fixed effects, and I have clarified this in the revised draft. Since the pandemic affected the entire country, a difference-in-differences approach would not be appropriate in this context, as there are no control states unaffected by the pandemic. Your suggestion is appreciated.

Given that this analysis focuses on a nationwide, aggregated measure of socioeconomic inequality, inter-state comparisons are not the objective. The dissimilarity index calculated in this study reflects inequality in the probability of receiving treatment at the national level. Although it would technically be possible to estimate the dissimilarity index for each state, this study avoids this due to limited sample sizes in some states, which would impede accurate calculation and exploration of determinants. I have discussed this issue in the revised manuscript.

>There is also a need for greater clarity on the contributions of the study relative to the existing literature, especially based on how you decide to address the above concern. It would be great to understand what the value add of this study is over the Rao Guthi, Visweswara et al. 2024 and Maiti et. Al, 2023 papers. Is the difference between the studies in terms of the samples (all states in the cited paper vs. during pandemic in the current study)? Or does the current study have methodological advantages over the existing ones?

(Reply)

Thank you for your suggestion. The strength of my study lies in the direct quantification of horizontal socioeconomic inequality in NCD treatment using the dissimilarity index and its decomposition analysis. This approach offers an advantage over traditional regression analysis by accounting for the entire distribution of treatment accessibility.

In the revised manuscript, I have included the following discussion to highlight the added value of this approach to the existing literature:

“First, it quantifies socioeconomic inequalities in the management of hypertension and diabetes using the dissimilarity index (D-index). Unlike a dichotomous regression approach, which estimates socioeconomic inequality based on the relationship between a binary treatment status and its determinants, the D-index reflects the overall distribution of the predicted probability of receiving treatment across the population. For example, if we are interested in disparities in hypertension treatment between educated and less-educated groups, traditional regression analysis would estimate the average difference in treatment rates between these two groups. While regression analysis might indicate that, on average, less-educated individuals have higher access to treatment than more-educated individuals, it does not reveal where within the distribution these disparities are most pronounced. In other words, regression provides a broad indication of inequality but does not show how treatment is distributed within each group. The dissimilarity index, on the other hand, captures absolute differences in treatment across the entire distribution by measuring the share of individuals who would need to switch treatment status to achieve equality between groups. Visual analysis of the D-index can further reveal whether disparities are pervasive across all levels or concentrated at specific levels, offering a more comprehensive view of the structure of socioeconomic inequality.”

I hope this revision clarifies the unique contributions of this approach.

---

## [Editor Report · Decision Letter 2]

20 Nov 2024

PONE-D-24-30611R2Socio-economic Gradients in Hypertension and Diabetes Management Amid the COVID-19 Pandemic in IndiaPLOS ONE

Dear Dr. Aizawa,

Thank you for re-submitting your manuscript to PLOS ONE after careful consideration of my comments. I feel that your study will be a valuable contribution but requires minor revisions before it fully meets PLOS ONE’s publication criteria. Therefore, I invite you to submit a revised version of the manuscript that addresses the points noted in the PDF.

Please submit your revised manuscript as soon as possible. If you will need more than usual time to complete your revisions, please reply to this message or contact the journal office at plosone@plos.org. Please include the following items when submitting your revised manuscript:A marked-up copy of your manuscript that highlights changes made to the original version. You should upload this as a separate file labeled 'Revised Manuscript with Track Changes'.An unmarked version of your revised paper without tracked changes. You should upload this as a separate file labeled 'Manuscript'.I look forward to receiving your revised manuscript.

Kind regards,

Bijetri Bose

Academic Editor

PLOS ONE
---

## [Author Response · Author response to Decision Letter 2]

29 Nov 2024

Response to editor’s comments:

Thank you for your valuable suggestions. I have addressed all the comments provided in my draft. Thanks to your feedback, the revise d draft has been significantly improved. My responses are summarized as follows.

>Please re-phrase or replace the hyphen with a comma; easy to mistake the hyphen as indicating a range.

(Reply)

I have changed it.

>The references you cite are mostly disease or country specific. Please consider the following references (and others if necessary) to better substantiate your claim:

Roy, Charlotte M., et al. "Assessing the indirect effects of COVID-19 on healthcare delivery, utilization and health outcomes: a scoping review." European Journal of Public Health 31.3 (2021): 634-640.

Moynihan, Ray, et al. "Impact of COVID-19 pandemic on utilisation of healthcare services: a systematic review." BMJ open 11.3 (2021): e045343.

(Reply)

Thank you for your advice. I have reorganized the references and incorporated the suggested literature.

>There are now several peer-reviewed studies to substantiate your claim. Please include a few of those.

(Reply)

I have incorporated relevant studies to strengthen my argument.

>Please include references for the supply side effects of the restrictions

(Reply)

I have incorporated relevant studies to strengthen my argument.

>Please check for accuracy. Sub-district hospitals also provide secondary care along with DHs.

(Reply)

Thank you for pointing that out. I have revised the sentence as follows: 

“At the secondary level, district hospitals (DHs), equipped with approximately 200 beds, cater to populations of 1--2 million and complement the CHCs in providing secondary care.”

>You reference a study from 2009. Please use an updated reference

(Reply)

Thank you for your suggestion. I have cited a more recent article.

>Please clarify who are the unit of analysis. Is it all adults over 18 yr for the hypertension & diabetes analysis but men and women of eligible ages for the medication analysis?

(Reply)

For clarity, I have added further explanation as follows: 

“Accordingly, all adults aged 18 years and older constitute the sample analysed in this study. For the medication analysis, only those aged 18 years and older who are identified as hypertensive or diabetic are included.”

>Please summarize the findings from Fig 2 and 3. If these do not provide any new information, please consider moving to the appendix.

(Reply)

I have moved them to the appendix as suggested.

>It will be helpful for readers if you briefly summarize the results from the pre-pandemic period.

(Reply)

Thank you for your suggestion. I’ve added a few sentences to summarize the results.

> Correct the title of Table 5

(Reply)

Thank you for pointing this out. I’ve corrected the title.

> Please consider adding a few lines on the comparison of results before and during the pandemic. This needs to be dealt with consistently throughout the paper since you are presenting results from both periods.

(Reply)

Thank you for your suggestion. I have included the following argument:

“Moreover, a comparison of the regression and decomposition results before and during the pandemic indicates that the worsening socioeconomic inequality in medication was closely associated with an increased wealth gradient. This underscores the greater importance of providing social support to less wealthy households requiring treatment to achieve horizontal equity in healthcare access during emergencies.”

---

## [Editor Report · Decision Letter 3]

3 Dec 2024

Socio-economic Gradients in Hypertension and Diabetes Management Amid the COVID-19 Pandemic in India

PONE-D-24-30611R3

Dear Dr. Aizawa,

We’re pleased to inform you that your manuscript has been judged scientifically suitable for publication and will be formally accepted for publication once it meets all outstanding technical requirements.

Kind regards,

Bijetri Bose

Academic Editor

PLOS ONE

---

## [Editor Report · Acceptance letter]

30 Dec 2024

PONE-D-24-30611R3 

PLOS ONE

Dear Dr. Aizawa, 

I'm pleased to inform you that your manuscript has been deemed suitable for publication in PLOS ONE. Congratulations! Your manuscript is now being handed over to our production team.

Kind regards, 

on behalf of

Dr. Bijetri Bose 

Academic Editor

PLOS ONE